# Ssn6 Interacts with Polar Tube Protein 2 and Transcriptional Repressor for RNA Polymerase II: Insight into Its Involvement in the Biological Process of Microsporidium *Nosema bombycis*

**DOI:** 10.3390/jof9100990

**Published:** 2023-10-05

**Authors:** Runpeng Wang, Yong Chen, Sheng Xu, Erjun Wei, Ping He, Qiang Wang, Yiling Zhang, Xudong Tang, Zhongyuan Shen

**Affiliations:** 1College of Biotechnology, Jiangsu University of Science and Technology, Zhenjiang 212100, China; qb1999517@126.com (R.W.); 18226746808@163.com (Y.C.); xu18609659561@163.com (S.X.); l723953668@163.com (E.W.); pinghe0306@163.com (P.H.); wqiang523@126.com (Q.W.); zhangyiling008@126.com (Y.Z.); xudongt@just.edu.cn (X.T.); 2Sericulture Research Institute, Chinese Academy of Agricultural Sciences, Zhenjiang 212100, China

**Keywords:** microsporidia, general transcriptional corepressor, polar tube protein, transcriptional repressor for RNA polymerase II

## Abstract

*Nosema bombycis* is a representative species of Microsporidia, and is the pathogen that causes pebrine disease in silkworms. In the process of infection, the polar tube of *N. bombycis* is injected into the host cells. During proliferation, *N. bombycis* recruits the mitochondria of host cells. The general transcriptional corepressor Ssn6 contains six tetratricopeptide repeats (TPR) and undertakes various important functions. In this study, we isolated and characterized *Nbssn6* of the microsporidium *N. bombycis*. The *Nbssn6* gene contains a complete ORF of 1182 bp in length that encodes a 393 amino acid polypeptide. Indirect immunofluorescence assay showed that the Ssn6 protein was mainly distributed in the cytoplasm and nucleus at the proliferative phase of *N. bombycis*. We revealed the interaction of Nbssn6 with polar tube protein 2 (Nbptp2) and the transcriptional repressor for RNA polymerase II (Nbtrrp2) by Co-IP and yeast two-hybrid assays. Results from RNA interference further confirmed that the transcriptional level of *Nbptp2* and *Nbtrrp2* was regulated by *Nbssn6*. These results suggest that Nbssn6 impacts the infection and proliferation of *N. bombycis* via interacting with the polar tube protein and transcriptional repressor for RNA polymerase II.

## 1. Introduction

Microsporidia are obligate intracellular pathogens related to fungi, and are obligate intracellular parasitic unicellular eukaryotes that can infect almost all animal species including important agricultural species [1,2,3,4], as well as humans, especially immunocompromised patients [5]. So far, more than 1500 species of Microsporidia belonging to 200 genera have been discovered [6,7,8]. *Nosema bombycis,* the first Microsporidium discovered in Europe in the mid-19th century, is the parasite that can trigger pebrine in silkworms [9,10]. Although Microsporidia are eukaryotes, they have significant differences from general eukaryotes. Microsporidia are highly reduced in morphology, subcellular structure, metabolism, and genome, but in terms of infection and proliferation, they have unique characteristics [11]. The differences of organelles between microsporidia and eukaryotes are significant. Microsporidia do not have the common organelles of general eukaryotes such as peroxisomes, Golgi bodies, etc., and their ribosome is more similar to prokaryotes than eukaryotes. Mitochondria in Microsporidia are degenerated, and the simplified mitochondria are called mitosomes [12,13]. The Tricarboxylic Acid Cycle (TCA) pathway has not yet been discovered in Microsporidia. Although all enzymes participating in glycolysis and part of the enzymes in the pentose phosphate pathway have been identified [14,15], in some Microsporidia such as *Enterocytozoon bieneusi, Enterocytozoon hepatopenaei,* and *Enterospora canceri*, parts of the enzymes participating in glycolysis were also lost [15]. Research has reported that Microsporidia have evolved strategies to maintain levels of ATP in the host, and provide additional nutrients for parasitism by stimulating the metabolic pathways [16].

Tetratricopeptide repeat (TPR) has been found in various proteins, and undertakes a variety of important functions. So far, more than 5000 tetratricopeptide repeat-containing proteins have been identified in diverse organisms through bioinformatic analysis [17]. We have screened the general transcriptional corepressor Ssn6 in the transcriptome of silkworm midgut infected with *N. bombycis* and the *N. bombycis* genome database (https://silkpathdb.swu.edu.cn/ accessed on 18 July 2022) [18]. The general transcriptional corepressor Ssn6 comprises six TPRs, and each repeat has 34 amino acids. TPR has been found in various proteins in different organisms, and plays important roles in many essential biological pathways, such as synaptic vesicle fusion, the targeting of peroxisome [2,19], the importing of mitochondria and chloroplast [20,21], as well as cell cycle control [22]. Tetratricopeptide motifs exist in the form of independent folds or part of protein folds. Ssn6 has been proved to interact with the transcriptional repressor Tup1 in other eukaryotic organisms [23,24], regulating a large number of genes by forming a corepressor and binding to target genes [25]. This Tup1-Ssn6 corepressor appears to be conserved from yeast to humans [26]. The Ssn6 mutant in *Saccharomyces cerevisiae* leads to the expression of glucose transport protein Hxt2p and up-regulation of the level of glucose transport [27]. In the yeast *Saccharomyces cerevisiae*, the Ssn6 (CYC8)-Tup1 protein complex represses the transcription of genes regulated by glucose, cell type, oxygen, DNA damage, and other signals [28]. However, the function of the Tup1-Ssn6 complex in Microsporidia remains unknown. In order to figure out whether Ssn6 acts as a transcriptional corepressor in Microsporidia, we screened the database and found that the Ssn6 is present in *N. bombycis*.

RNA polymerase II has the ability of RNA transcript elongation and proofreading [29], and is the core of the complex apparatus that is responsible for the regulated synthesis of mRNA [30]. Many reports suggest that RNA polymerase II may mediate alterations in chromatin structure and play a key role in the regulatory mechanisms that influence transcriptional initiation, RNA chain elongation, RNA processing, and transcription termination [31]. Transcriptional repression is an important component of the regulatory networks that govern gene expression, and RNA polymerase II was proved to be impeded by immediate early protein 2, Tup1, lac repressor, and so on [32,33,34]. A kind of protein annotated as a transcriptional repressor for RNA polymerase II (Trrp2) of Microsporidia has been screened in the database. Trrp2 owns a domain named SCOP_d1xgra, which contains seven WD40 repeats and has been found in transcriptional repressors of the Groucho/Transducin-like Enhancer of Split (TLE) family in invertebrates and vertebrates [35]. In mammalian cells, TLE was proved to have apparent homology with the C-terminal WD-repeat region in Tup1 and could interact with Ssn6 [36,37]. This evidence suggests that Trrp2 may have a similar function to Tup1.

In this study, we successfully cloned the *Ssn6* gene (Nbssn6) and *Nbtrrp2* gene (transcriptional repressor for RNA polymerase II) of *N. bombycis*. Domain analysis showed that the Ssn6 of *N. bombycis* has six TPR motifs, and the Trrp2 has a coiled coil region and a SCOP d1gxra domain. The Nbssn6 recombinant protein was expressed by a prokaryotic expression vector. Indirect immunofluorescence assay showed that Ssn6 was mainly distributed in the cytoplasm and nuclei of *N. bombycis*. Yeast two-hybrid assay confirmed the interaction between Ssn6 and polar tube protein 2 and Nbtrrp2, and silencing of the *Nbssn6* gene by RNAi resulted in down-regulated expression of the *Nbptp2* gene and *Nbtrrp2* gene.

## 2. Materials and Methods

### 2.1. Parasite and Host

*N. bombycis* was preserved at the Institute of Sericulture, Chinese Academy of Agricultural Sciences. The silkworm strain P50 and the BmN cell line were used in the experiments. The BmN cell was isolated from silkworms and kept in our laboratory.

### 2.2. Cloning and Bioinformatic Analysis of Nbssn6 and Nbtrrp2 Genes

The purified spore suspension of *N. bombycis* (10^9^ spores/mL) was crushed in a bead grinder for 1 min with 1:1 acid-washed glass beads, then cooled on ice for 2 min, repeated 6 times. The genomic DNA was extracted using a fungal genomic DNA extraction kit (Sangon Biotech, Shanghai, China), and the DNA solution was kept at −20 °C after concentration measurements.

The nucleotide sequences of *Nbssn6* (GenBank: EOB14249.1) and *Nbtrrp2* (GenBank: EOB11319.1) were obtained from the NCBI database (https://www.ncbi.nlm.nih.gov, accessed on 15 May 2022). The specific primers were synthesized by Sangon Biotech, Shanghai (Table 1). The PCR amplification system comprised PrimeSTAR HS DNA Polymerase (R010Q, Takara, Shiga, Japan) 25 μL, forward and reverse primers 2 μL(10 mM), genomic DNA 2 μL (100 ng), and ddH_2_O to the total volume of 50 μL. The PCR products were separated by 1% agarose gel and isolated with an Axyprep DNAGel Extraction Kit (AP-GX-4, Axygen bioscience, Glendale, AZ, USA) before being cloned into the pMD19-T-Vector (6013, Takara Biotechnology, Dalian) with a polyA tail (RR014A, Takara Biotechnology, Dalian, China). The pMD19-T-*Nbssn6* vector was transformed into *E. coli* TOP10 competent cells, then was cultured on ampicillin/IPTG/X-Gal agar plates.

The molecular weight and isoelectric point of Nbssn6 and Nbtrrp2 protein were predicted by Compute pI/Mw tool (https://web.expasy.org/cgi-bin/compute_pi/pi_tool, accessed on 15 May 2022) [38]. The signal peptide was predicted by SignalP-5 (http://www.cbs.dtu.dk/services/SignalP/, accessed on 16 May 2022) [39]. The phosphorylation and glycosylation sites were predicted by DTU Health Tech (http://cello.life.nctu.edu.tw/, accessed on 20 May 2022) [18]. A schematic map of protein domains was made by SMART (http://smart.embl-heidelberg.de/, accessed on 15 May 2022) [40,41]. The three-dimensional structure of protein was predicted by the AlphaFold Protein Structure Database (https://alphafold.ebi.ac.uk/, accessed on 9 September 2023) [42].

### 2.3. Expression and Purification of Nbssn6 Recombinant Protein

The correctly sequenced pMD19-T-*Nbssn6* positive plasmid and pET-28a empty plasmid were identified by double enzyme digestion. The product was connected to the vector and transformed into *E. coli* BL21 Star (DE3) (B528419, Sangon Biotech, Shanghai, China), and the correctly identified positive colonies were used for prokaryotic expression.

The overnight-cultivated bacteria were added into LB liquid medium (Kana) at 1:100 and incubated in a shaker at 37 °C until the OD reached 0.6. Before induction, 1 mL of the bacterial cells was taken as a negative control. The remaining bacterial cells were induced with 0.5 mM IPTG for 12 h at 20 °C, then resuspended in PBS after removing the supernatant by centrifugation. After ultrasonic disruption (300 W, 10 s, 10-s intervals, repeated 30 times), the supernatant and precipitate were collected separately, and analyzed by SDS-PAGE.

The pET-28a-*Nbssn6 E. coli* was disrupted by sonication. After centrifugation, the precipitate was dissolved with urea, and the sample was loaded on a pre-equilibrated NTA purification column, which was successively washed with 10 volumes of NTAU-0 buffer, and 1 volume of NTAU-X buffer. Different components were collected step by step and detected by SDS-PAGE. Finally, the purified recombinant protein was obtained.

### 2.4. Preparation of Nbssn6 Polyclonal Antibodies and Western Blot Analysis

Two white New Zealand rabbits were immunized with 1 mL of the Nbssn6 recombinant protein mixed with 1 mL of Freund’s complete adjuvant (Sigma, Darmstadt, Germany) on the first day. On the 15th, 29th, and 43rd days, the recombinant protein mixed with Freund’s incomplete adjuvant (Sigma, Darmstadt, Germany) was used for subsequent injections. Finally, blood was collected from the carotid artery of each rabbit on the 53rd day, and incubated overnight at 4 °C. After centrifugation (4 °C, 10,000 rpm) for 30 min, the supernatant was collected. The antigen affinity column was prepared by connecting the antigen protein to CNBr-activated Sephrose 4B (C9142, Merck, Darmstadt, Germany). The serum was purified by the antigen affinity column. The specific polyclonal antibodies against the target protein were finally obtained, and the antibody titer was detected by ELISA. Pre-immune serum fom the rabbits was used as a negative control.

The total protein of the *N. bombycis* was extracted as follows: 1 mL Percoll purified spore suspension of *N. bombycis* was centrifuged to remove the supernatant, and resuspended in 1 mL NP-40 lysate with 1:1 acid-washed glass beads, then crushed with a tissue grinder (OSE-TH-01, TIANGEN, Beijing, China) for 1 min, and cooled on ice for 3 min, repeated 6 times. The lysed suspension was centrifuged at 12,000 g for 10 min at 4 °C, and the supernatant was obtained as the total protein of *N. bombycis.*

Western blot was performed using Nbssn6 polyclonal antibodies purified by affinity chromatography as the first antibody and the HRP-conjugated goat anti-rabbit IgG antibody (D110058, Sangon Biotech, Shanghai, China) as the second antibody at room temperature for 1 h. Pre-immune serum was used as the negative control. The PVDF membrane was incubated with Tanon™ High-sig ECL Western blot substrate for 2 min, imaged under a Tanon 5200 Multi imaging system.

### 2.5. Immunolocalization of Nbssn6 in N. Bombycis

In order to investigate the subcellular localization of Nbssn6 in the proliferative phase of *N. bombycis* in BmN cells, spore suspensions of *N. bombycis* and 0.2 M KOH solution were preheated at 27 °C for 2 h, respectively, then mixed 1:1 (volume) to make the final concentration of 0.1 M KOH. The mixture was incubated at 27 °C for 40 min to germinate the spores. The suspension of germinating spores was slowly dropped into the BmN cells. After incubating for 1 h, the culture medium free of serum was replaced with TC-100 insect culture medium containing 10% serum. After the BmN cells were cultured overnight, the culture medium was exchanged with fresh culture medium the next day. The BmN cells infected with *N. bombycis* were added into a 6-well plate with coverslip at different days post-infection. After washing three times with PBST, the *N. bombycis-* infected BmN cells were fixed with 4% paraformaldehyde. The polyclonal antibody against Nbssn6 (1.64 mg/mL) was labeled with Alexa Fluor 488-conjugated goat anti-rabbit IgG (D110061, Sangon Biotech, Shanghai, China), while the polyclonal antibody against Nb-actin (0.5 mg/mL) was labeled with Cy5- conjugated goat anti-rabbit IgG (D110071, Sangon Biotech, Shanghai). The immunolocalization was observed under a fluorescence inverted microscope (IX-71, Olympus, Tokyo, Japan) [43].

### 2.6. CO-IP Analyses

All of the *N. bombycis* proteins were extracted as mentioned above. A total of 2 μg of the Nbssn6 polyclonal antibody and 5 μL of the 100x protease inhibitor cocktail (Beyotime, Shanghai) were added into a 500 μL protein solution of *N. bombycis*. After incubation at 4 °C for 16 h, 50 μL of Protein A Magnetic Beads (P2102, Beyotime, Shanghai, China) were added for further incubation at 4 °C for 6~8 h. After the supernatant was removed, the magnetic beads were washed with 1 mL PBS 3 times, then resuspended with 50 μL 1× SDS-PAGE Sample Loading Buffer. After being treated at 100 °C for 10 min, the magnetic beads were removed with a Magnetic Separation Rack (FMS012, Beyotime, Shanghai, China). The same procedure was performed as a negative control, except that the Nbssn6 polyclonal antibody was replaced by rabbit IgG. The CO-IP samples were detected by SDS-PAGE and Coomassie analysis, and the specific bands were cut for mass spectrometry analysis.

### 2.7. Yeast Two-Hybrid Assay

*Nbssn6, Nbtrrp2,* and *Nbptp2* (polar tube protein 2 of *N. bombycis*) were amplified from genomic DNA of *N. bombycis* using specific primers (Table 1). *Nbtrrp2* and *Nbptp2* were ligated into the bait vector pGBKT7, while Nbssn6 was ligated into the yeast two-hybrid prey vector pGADT7-BK using an In-Fusion HD cloning kit (638909, TaKaRa, Shiga, Japan). The bait recombinant vector and prey recombinant vector were co-transformed into Y2H Gold yeast cells (YC1002, Weidi Biotech, Shanghai, China), then cultured on SD−Leu/−Trp plates (630489, TaKaRa, Shiga, Japan) at 30 °C. A single colony on SD−Leu/−Trp plates was selected and cultured on SD−Ade/−His/−Leu/−Trp plates containing X-α-gal (630462, TaKaRa, Shiga, Japan) at 30 °C.

### 2.8. RNAi and RT-qPCR

Forty newly molted silkworm larvae of the 5th instar were fed mulberry leaves smeared with 20 mL of *N. bombycis* spores (10^8^ spores/mL) for 6 h. Either 3 μL of siRNA (1 μg) or a nonsense fragment was injected into each infected silkworm. Midgut tissues of 10 silkworm larvae were taken at 24 h, 48 h, 72 h, 96 h, and 120 h post injection, respectively, then washed with PBS and stored at −80 °C.

The midgut tissue was lysed with 1 mL of RNAiso plus lysate (9108Q, Takara, Dalian). The total RNA of the midgut tissue was extracted with a Mini BEST Universal RNA Extraction Kit (9767, Takara, Shiga, Japan). The cDNA was synthesized with PrimerScript^®^ RT Master Mix (RR036B, Takara, Shiga, Japan). RT-qPCR was performed according to the TB GreenTM Premix ExTaqTM II (Tli RNase H Plus) (RR820A, Takara, Shiga, Japan) kit manufacturer’s instructions, conducted at 95 °C for 30 s, followed by 40 cycles of 95 °C for 5 s and 60 °C for 30 s on QuantStudio 3 (ABI, Waltham, MA, USA). The *β-tubulin* gene of *N. bombycis* was used as a reference gene, and the primers were listed in Table 1. Three independent repeated experiments were performed. The transcription levels were calculated by the 2^-∆∆ct^ method. GraphPad Prism 8.0 (GraphPad Software, San Diego, CA, United States) was used to conduct the multiple t tests. 

In order to analyze the effect of the knockdown of *Nbssn6* on the proliferation of *N. bombycis*, specific primers Nbβ-tubulin-qF and Nb β-tubulin-qR (Table 1) were designed to detect the copy number of Nbβ-tubulin. The standard template was prepared according to the method of the previous report [44], and the standard curve covered five orders of magnitude (5.6 × 10^2^–10^6^). The multiple t tests were conducted using GraphPad Prism 8.0 with three biological replicates.

## 3. Results

### 3.1. Cloning and Sequence Analysis of Nbssn6 and Nbtrrp2 Genes

The *Nbssn6* gene contains a complete 1182 bp ORF that encodes a 393 amino acid polypeptide (Figure 1A). Nbssn6 has a predicted pI of 7.99, and a predicted molecular weight of 46.4 kDa. The Nbssn6 has 30 phosphorylation sites, and no signal peptide and glycosylation site. Secondary structure analysis showed that the alpha-helix, random coils, beta sheets, and extended fragment account for 58.78%, 25.95%, 5.34%, and 9.92%, respectively. Domains predicted by NCBI showed that Nbssn6 has six TPRs which are located at 50-78 aa, 119-152 aa, 156-189 aa, 190-218 aa, 258-291 aa, and 292-325 aa in *N. bombycis* (Figure 2B). These TPR domains were also shown in three-dimensional model. Each TPR domain was predicted to form a helix-turn-helix arrangement V-shaped structure (Figure 2A). Multiple sequence alignment showed that Nbssn6 shared a certain degree of homology with that of other microsporidian species such as *Nosema granulosis, Nosema ceranae,* and *Encephalitozoon cuniculi*. In particular, the homology with *N. granulosis* was 51.56% (Figure 2B).

The *Nbtrrp2* gene contains a complete 1278 bp ORF that encodes a 425 amino acid polypeptide (Figure 1B). Nbtrrp2 has a predicted pI of 8.43, and a predicted molecular weight of 48.5 kDa. Domain analysis shows that Trrp2 contains a SCOP d1gxra domain which is located at 62-293 aa and forms a toroidal structure based on seven blades. Each blade consists of a four-stranded antiparallel β sheet (Figure 3A). The SCOP d1gxra domain shares homology with the human Groucho/TLE1 transcriptional corepressor, and is very similar to Tup1 β propellers [36] (Figure 3B). Multiple sequence alignment showed that Nbtrrp2 shares a certain degree of homology with that of other microsporidian species at N-terminal. The homology of *N. bombycis* with *N. granulosis Encephalitozoon romaleae*, and *Encephalitozoon hellem* were 41.50%, 36.55% and 33.10%, respectively (Figure 3C).

### 3.2. Expression and Western Blot Analysis of Nbssn6 Protein

SDS-PAGE results showed that a protein band with an expected size of 49 kDa is mainly expressed in the form of inclusion bodies (Figure 4A). The results of NTAU purification showed that the target protein was obtained by elution with 200 or 500 mM imidazole (Figure 4B). 

The prepared antibody showed high sensitivity and specificity against Nbssn6 through titer testing and Western blot analysis. Western blot analysis was performed by using the total protein of *N. bombycis* and the Nbssn6 recombinant protein, with pre-immune serum as the negative control. The results showed that specific bands were detected both in the recombinant protein and total protein of *N. bombycis* (Figure 4C). The pre-immune serums did not detect the target protein (Figure 4D). These results indicated that the Nbssn6 polyclonal antibody has a specific antigen–antibody reaction and can be used in subsequent experiments.

### 3.3. Subcellular Localization of Nbssn6 Protein in N. Bombycis

The localization of Nbssn6 and its co-localization with Nb-actin in the intracellular proliferative phase of *N. bombycis* were performed by indirect immunofluorescence assay (IFA). The Nbssn6 antibody was coupled with Alexa Fluor 488, while the Nb-actin antibody was coupled with Cy5, and the nucleus was stained with DAPI, respectively. Pre-immune serum was used as a negative control. The IFA results showed that during the proliferative phase, Nbssn6 was mainly distributed in the perinuclear cytoplasm (Figure 5A) or in most of the cytoplasm (Figure 5B), and was sometimes located in the nuclei (Figure 5C). No green fluorescence could be found in negative control (Figure 5D). In order to further explore the distribution of Nbssn6 in the proliferative phase, colocalization of Nbssn6 was performed with Nb-actin, which is a kind of multifunctional protein that can form microfilaments and is an indispensable part of the cytoskeleton of *N. bombycis* [23]. During the early proliferative phase, Nb-actin (red fluorescence) was evenly distributed in the cytoplasm, while Nbssn6 (green fluorescence) was mainly distributed in the nuclei (Figure 6A) or the whole cell (Figure 6B). During the late proliferative phase, we found that Nbssn6 was distributed in both the nuclei and perinuclear cytoplasm (Figure 6C). No green fluorescence was found in the negative control (Figure 6D). These results indicated that Nbssn6 is distributed in the cytoplasm and nuclei, and might shuttle between the nuclei and cytoplasm.

### 3.4. Co-Immunoprecipitation and Mass Spectrometry Analysis

Two specific bands from Co-IP samples were detected by SDS-PAGE (Figure 7), and were subsequently sequenced by mass spectrometry analysis. The probable interacting proteins are listed in Table 2. Some crucial proteins, such as polar tube proteins, the T-complex protein family, and heat shock proteins were screened as candidates for further study.

### 3.5. Interaction between Nbssn6 and Nbtrrp2 and Knockdown of the Nbssn6 Gene Down-regulated the Transcription Level of the Nbtrrp2 Gene

The positive and negative control settings are shown in Figure 8A. A yeast two-hybrid assay showed that the yeast with plasmids of activating domain (AD)–*Nbssn6* and binding domain (BK)–*Nbtrrp2* grew successfully on synthetic dropout (SD)−Ade/−His/−Leu/−Trp selective medium with 5-Bromo-4-chloro-3-indoxyl-α-d-galactopyranoside (X-α-gal) (Figure 8B), suggesting that Nbssn6 interacts with Nbtrrp2.

After RNA interference, the transcription level of the *Nbssn6* and *Nbtrrp2* genes was detected by RT-qPCR using the *β-tubulin* as reference gene. The relative transcription level of the *Nbssn6* gene was extremely significantly down-regulated at all time points (*p* < 0.01) (Figure 9A), while the relative transcription level of the *Nbtrrp2* gene was significantly down-regulated at all time points (*p* < 0.01), especially at 24 h, 48 h, and 72 h (Figure 9B).

### 3.6. Interaction between Nbssn6 and Nbptp2 and the Knockdown of the Nbssn6 Gene Down-regulated the Transcription Level of the Nbptp2 Gene

Nbptp2 was predicted to have a signal peptide containing 26 amino acids and a PTP2 domain (Figure 10A). The positive and negative control settings are shown in Figure 10B. The yeast two-hybrid assay showed that the yeast with plasmids of activating domain (AD)–*Nbssn6* and binding domain (BK)–*Nbptp2* grew successfully on synthetic dropout (SD)−Ade/−His/−Leu/−Trp selective medium with 5-Bromo-4-chloro-3-indoxyl-α-d-galactopyranoside (X-α-gal) (Figure 10C), indicating that Nbssn6 protein interacts with Nbptp2.

After the knockdown of *Nbssn6* by RNA interference, the relative transcription level of *Nbptp2* was extremely significantly down-regulated at 24 h, 48 h, and 72 h (*p* < 0.001), then extremely significantly up-regulated at 96 h (Figure 11), suggesting that the transcription level of *Nbptp2* was dependent on *Nbssn6*, and that the silencing of *Nbssn6* had a significant impact on *Nbptp2*.

### 3.7. Knockdown of the Nbssn6 Gene Down-regulated the Proliferation Level of N. Bombycis

The standard curve was established by calculating plasmid pMD19-T-*Nbβ-tubulin* gradient dilution samples. The proliferation level was indicated based on the copy number of the housekeeping gene *Nbβ-tubulin* at different time points of the NC and RNAi samples. The proliferation was significantly down-regulated at 24 h, and extremely significantly down-regulated at 72 h and 96 h. These results indicate that Ssn6 is involved in the infection and proliferation of *N. bombycis* (Figure 12).

## 4. Discussion

The tetratricopeptide repeats (TPR) play crucial roles in the functions of Ssn6, such as interacting with Tup1 [24]. Adjacent TPR units in turn form an overall supercoiled structure, and the supercoil forms a pair of concave and convex surfaces that allow multiple ligands to bind. Generally, the ligands bind to the concave surface of the tetratricopeptide repeat surface [17,45]. Many proteins containing TPR have crucial functions and are involved in various biological processes. TPR in the adaptor protein LGN is the structural basis for the recognition of the scaffold protein Frmpd4/Preso1 [46]. The histone chaperone sNASP binds to a conserved peptide motif within the globular core of histone H3 through TPR [47], and more functions of TPR are to be discovered. In *Saccharomyces cerevisiae*, Ssn6 and Tup1 protein form a corepressor complex, repressing the transcription of diverse genes [48]. A lack of Ssn6 in *Candida albicans* leads to mass switching from the white to the opaque cell type [49].

Thus far, research about TPR domain-containing proteins in Microsporidia are rare. In *E. cuniculi*, the E3 ubiquitin ligase APC, which degrades specific cell cycle regulatory proteins, is a sub-complex composed of the TPR subunits Cdc16, Cdc23, and Cdc27 [50]. In this study, we searched and blasted homologous proteins of Ssn6 and Tup1 in Microsporidia, and found homologous sequences of Ssn6 in some Microsporidia such as *Encephalitozoon cuniculi*, *N. bombycis*, *N. granulosis,* and *N. ceranae*. Whereas the Tup1 homologous proteins with the Tup_N domain only exist in the *Amphiamblys* sp. (Metchnikovellidae), a group of unusual Microsporidia were among the earliest-branching lineages in Microsporidia, which shows a distant genetic relationship with other Microsporidia [51,52]. Interestingly, we found the transcriptional repressor for RNA polymerase II (Trrp2) in some Microsporidia such as *N. bombycis*, *N. granulosis, En. hellem,* and *En. romaleae*. A yeast two-hybrid assay showed the interaction of Nbssn6 and Nbtrrp2. RNAi indicated that a knockdown of *Nbssn6* significantly down-regulated the transcription level of *Nbtrrp2*, further proving the functional relationship between Ssn6 and Trrp2 in *N. bombycis*. The Trrp2 of Microsporidia contains a SCOP d1gxra domain that has homology with the human Groucho/TLE1 transcriptional corepressor, which regulates the expression of a variety of genes and is involved in numerous developmental processes [53]. The corepressor TOPLESS (TPL) in *S. cerevisiae* is a member of the larger pan-eukaryotic Tup1/TLE/Groucho corepressor family, and showed activity in a synthetic transcriptional repression [54]. Taken together, we speculate that the Trrp2 of Microsporidia possesses a similar function as Tup1, but the mechanism needs further research.

Microsporidia own a unique process during infection, represented by the sudden extrusion of the polar tube for initiating entry of the parasite into new host cells [55]. So far, there are six polar tube proteins identified and 10 potential polar tube proteins screened in *N. bombycis* [56,57]. Ptp2 is a structural protein with a basic lysine-rich core and an acidic tail, and plays a major role in the polar tube assembly of Microsporidia [55]. The Ptp1-Ptp2 polymer has been proved to have a close connection with polar tube ejection during the infection process [58,59]. However, the synthesis, transportation, arrangement, and orderly orientation of polar tube proteins in *N. bombycis* remain unknown [60]. In this study, the Nbssn6 interacts with Nbptp2, and RNAi results show that the expression level of *Nbptp2* is extremely significantly down-regulated at the early and middle proliferative phases of *N. bombycis*, suggesting that Nbssn6 may be involved in the infection of *N. bombycis* via interacting with Nbptp2.

## Figures and Tables

**Figure 1 jof-09-00990-f001:**
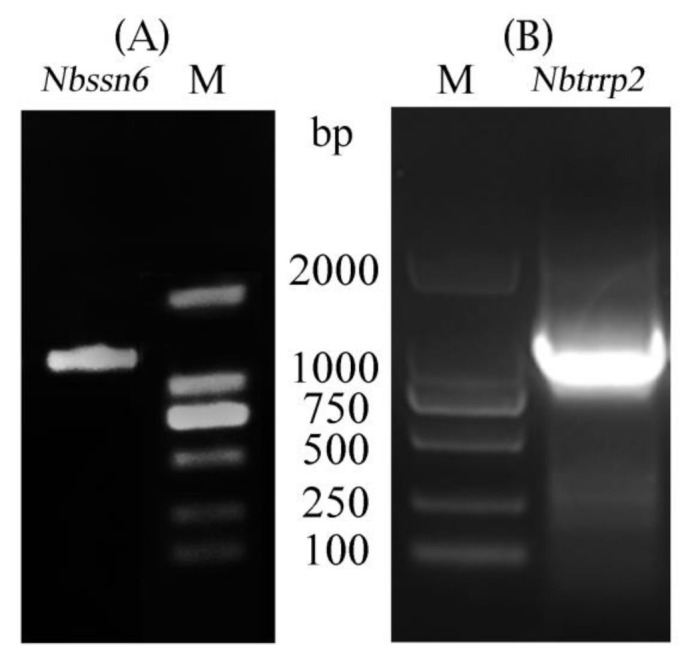
Cloning of *Nbssn6* and *Nbtrrp2* genes. (**A**) DNA agarose gel electrophoresis of *Nbssn6*. (**B**) DNA agarose gel electrophoresis of *Nbtrrp2*.

**Figure 2 jof-09-00990-f002:**
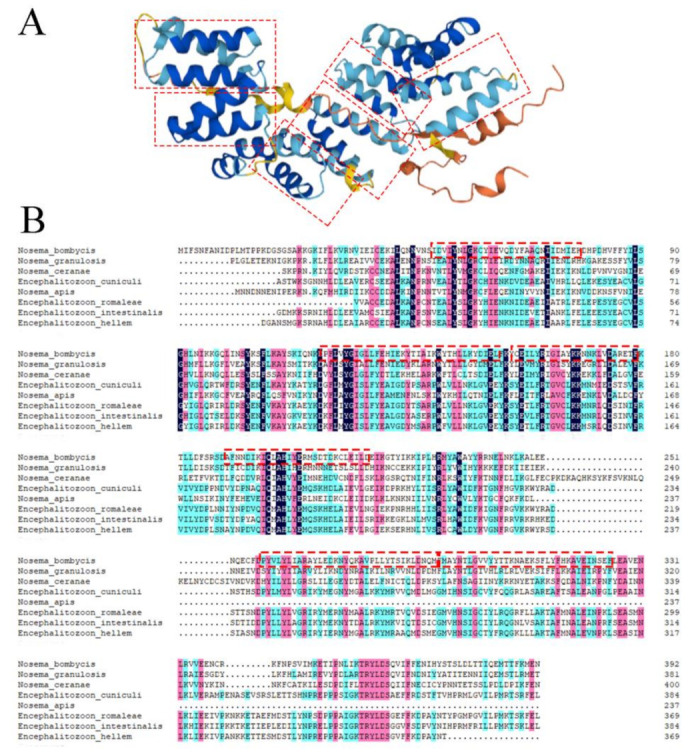
Sequence analysis of the *Nbssn6* gene. (**A**) Three-dimensional model of Ssn6. (**B**) Amino acid alignment of Ssn6 from *Nosema bombycis* and other microsporidian species. Red box: TPR domains predicted by NCBI.

**Figure 3 jof-09-00990-f003:**
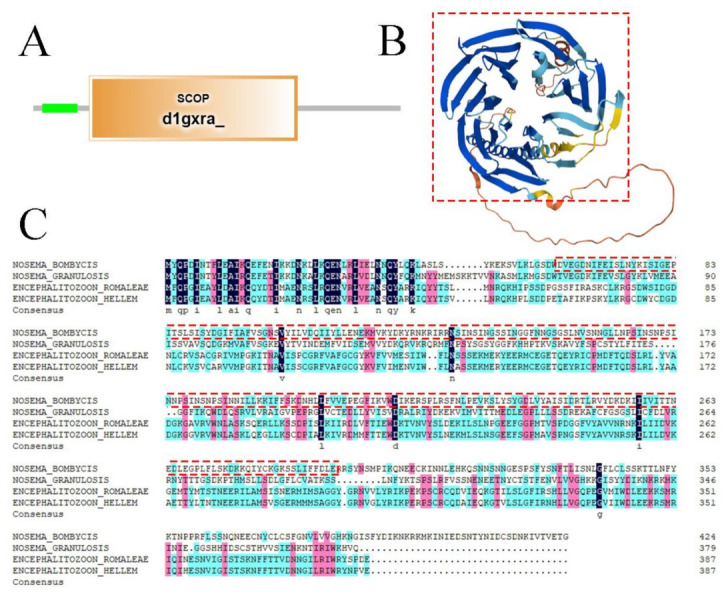
Sequence analysis of the *Nbtrrp2* gene. (**A**) Domains of Nbtrrp2. (**B**) Three-dimensional models of Nbtrrp2. (**C**) Amino acid alignment of Trrp2 from *Nosema bombycis* and other microsporidian species. Red box: SCOP_d1xgra domain.

**Figure 4 jof-09-00990-f004:**
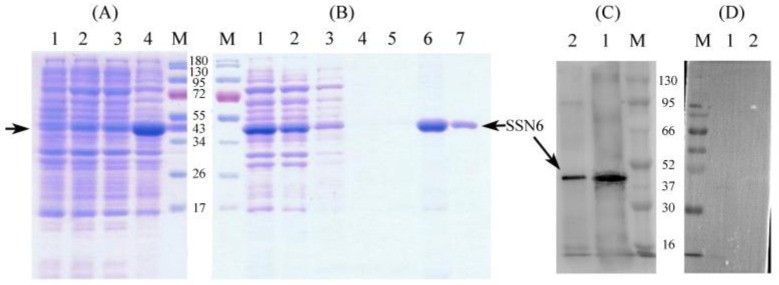
Expression, purification, and Western blot analysis of Nbssn6 protein. (**A**) SDS-PAGE analysis of expressed recombinant protein. Lane M: Protein molecular weight marker. Lane 1: Recombinant bacterial lysate (uninduced). Lane 2: Recombinant bacterial lysate induced with 0.5mM IPTG. Lane 3: Ultrasound supernatant. Lane 4: Ultrasonic precipitation. Black arrow: Ssn6 (**B**) SDS-PAGE analysis of protein purification. Lane M: Protein molecular weight marker. Lane 1: Urea solution for ultrasonic precipitation (sample to be purified). Lane 2: Flow through. Lane 3: NTAU-10 elution. Lane 4: NTAU-20 elution. Lane 5: NTAU-50 elution. Lane 6: NTAU-200 elution. Lane 7: NTAU-500 elution. (**C**) Western blot of Nbssn6 protein. Lane M: Protein molecular weight marker. Lane 1: Recombinant protein. Lane 2: Total protein of *N. bombycis*. (**D**) Pre-immune serum control. Lane M: Protein molecular weight marker. Lane 1: Recombinant protein. Lane 2: Total protein of *N. bombycis*.

**Figure 5 jof-09-00990-f005:**
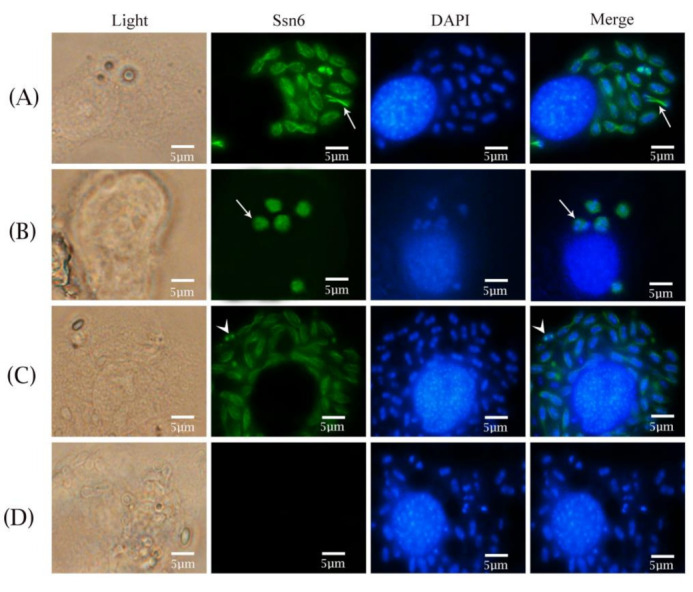
Subcellular localization of Nbssn6 protein. The Nbssn6 antibody was coupled with Alexa Fluor 488 (green). DAPI (blue) was used to stain the nuclei of the host cells and *N. bombycis*. (**A**) White arrow indicated that Nbssn6 was mainly distributed in the perinuclear cytoplasm; (**B**) White arrow indicated that Nbssn6 was mainly distributed in the cytoplasm; (**C**) White arrow head indicated that Nbssn6 was distributed in the nuclei; (**D**) Negative control. Scale bars, 5 μm.

**Figure 6 jof-09-00990-f006:**
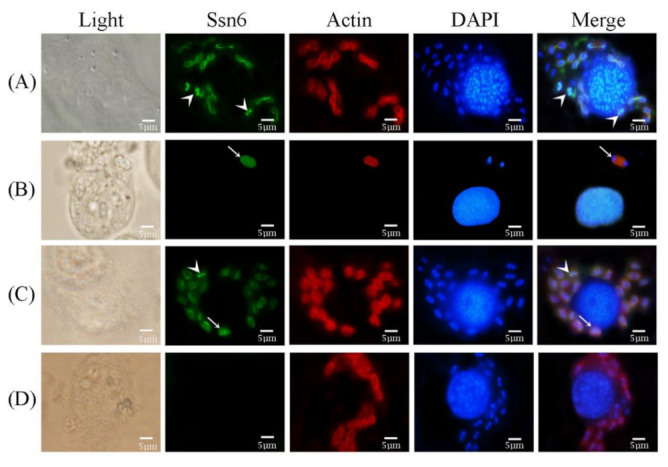
Colocalization of Nbssn6 with Nb-actin. The Nbssn6 antibody was coupled with Alexa Fluor 488 (green). The Nb-actin antibody coupled with Cy5 (red) were used to label the actin of *N. bombycis*. DAPI (blue) was used to stain the nuclei of the host cells and *N. bombycis*. (**A**) White arrow head indicated that Nbssn6 was distributed in the nuclei; (**B**) White arrow indicated that Nbssn6 was evenly distributed in *N. bombycis*; (**C**) White arrow indicated that Nbssn6 was distributed in both the nuclei and perinuclear cytoplasm, while the white arrow head indicated that Nbssn6 was distributed in the nuclei; (**D**) Negative control. Scale bars, 5 μm.

**Figure 7 jof-09-00990-f007:**
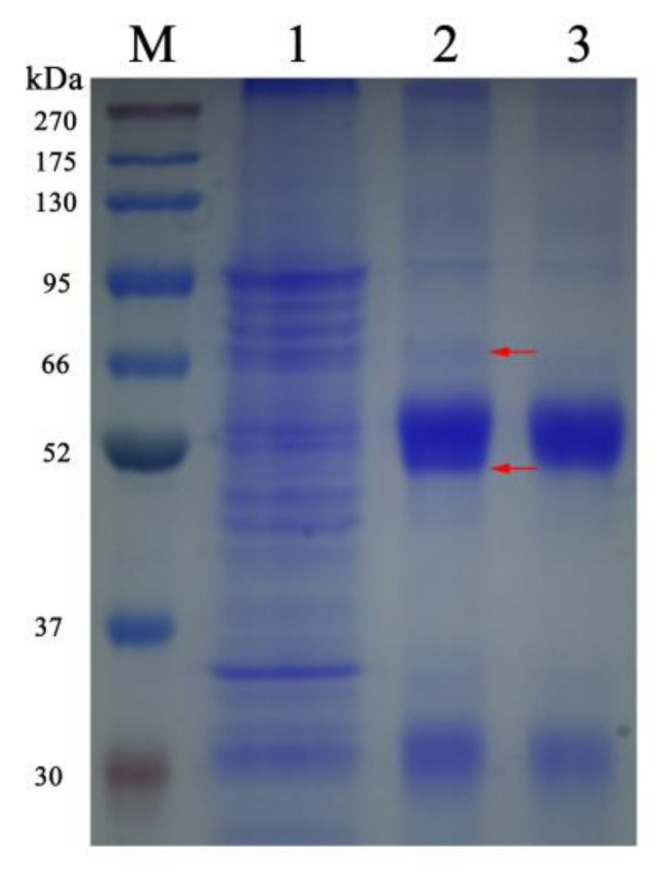
SDS-PAGE analysis of Co-IP results. Lane M: Protein molecular weight marker. Lane 1: Total protein of *N. bombycis*. Lane 2: Co-IP result using Nbssn6 antibody. Lane 3: Co-IP result using rabbit IgG antibody. Red arrow: specific band.

**Figure 8 jof-09-00990-f008:**
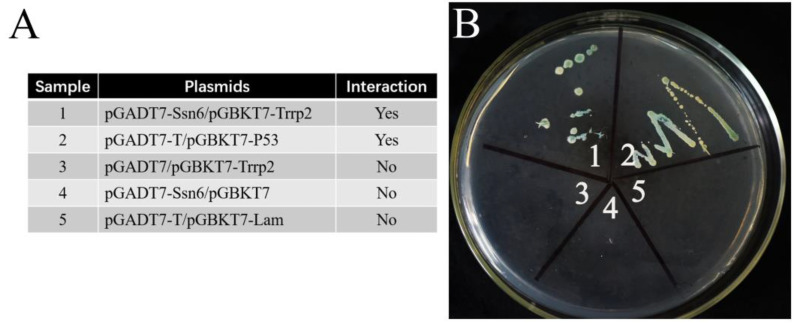
Interaction between Nbssn6 and Nbtrrp2. (**A**) Explanation for the 5 areas on the culture medium: the interaction of pGADT7-T and pGBKT7-P53 was used as the positive control, and the interaction of pGADT7-T and pGBKT7-lam was used as the negative control. (**B**) Yeast two-hybrid assay of the interaction between Nbssn6 and Nbtrrp2.

**Figure 9 jof-09-00990-f009:**
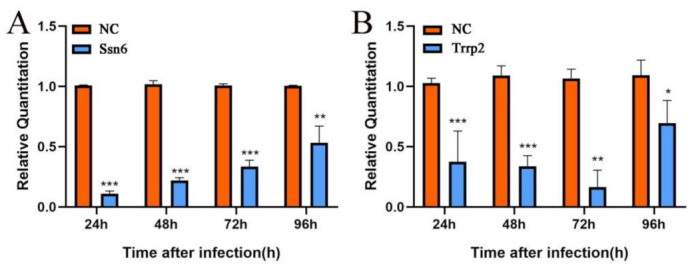
Effect of the knockdown of *Nbssn6* on the transcription level of *Nbtrrp2*. (**A**) Effect of RNAi on the transcription level of the *Nbssn6* gene. (**B**) Effect of RNAi on the transcription level of the *Nbtrrp2* gene. Error bars represent the standard deviations of 3 independent replicates (*n* = 3, mean ± SE, * *p* < 0.05, ** *p* < 0.01, *** *p* < 0.001, NC, β-tubulin).

**Figure 10 jof-09-00990-f010:**
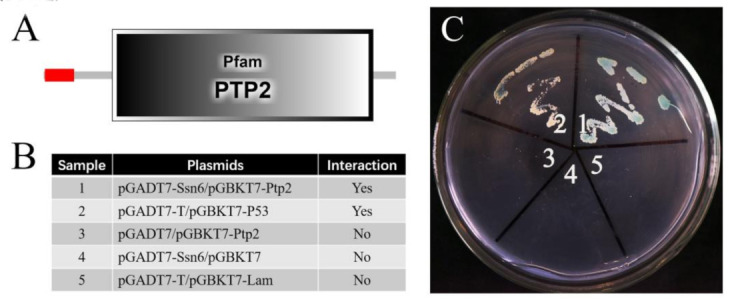
Interaction between Nbssn6 and Nbptp2. (**A**) Domains of Nbptp2. (**B**) Explanation for the 5 areas on the culture medium: the interaction of pGADT7-T and pGBKT7-P53 was used as the positive control, and the interaction of pGADT7-T and pGBKT7-lam was used as the negative control. (**C**) Yeast two-hybrid assay of the interaction between Nbssn6 and Nbptp2.

**Figure 11 jof-09-00990-f011:**
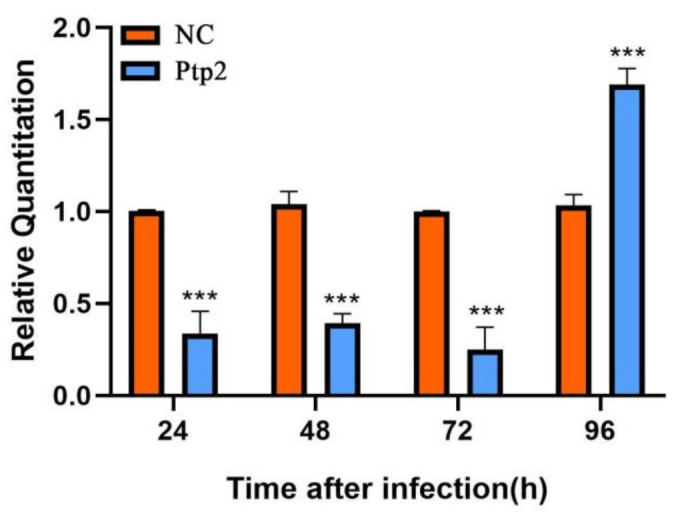
The effect of the knockdown of *Nbssn6* on the transcription level of *Nbptp2*. Error bars represent the standard deviations of 3 independent replicates (*n* = 3, mean ± SE, *** *p* < 0.001, NC, β-tubulin).

**Figure 12 jof-09-00990-f012:**
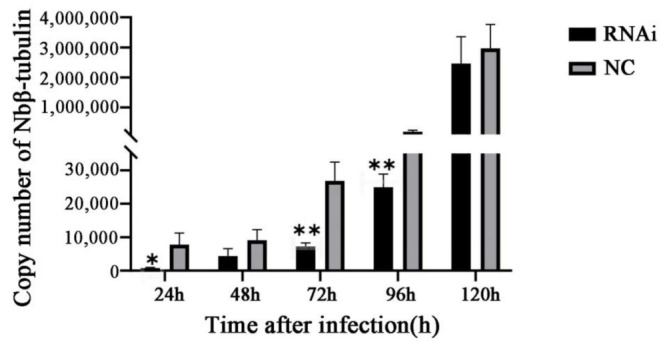
Effect of down-regulating the *Nbssn6* gene on *N. bombycis* proliferation via RNAi. The proliferation level is obtained by calculating the copy number of *Nbβ-tubulin*. Error bars represent the standard deviations of 3 independent replicates (*n* = 3, mean ± SE, * *p* < 0.05, ** *p* < 0.01, NC, β-tubulin).

**Table 1 jof-09-00990-t001:** Primer sequences for PCR, yeast two-hybrid, RNAi and RT-qPCR.

Gene	Sense/ Antisense	Sequences
**Nbssn6 (PCR)**	Forward	GAATTCATGATTTTTTCTAATTTCGCAAATATTGATCCTTTAATGA (5′-3′)
	Reverse	GTCGACTCACAAATTTTCCATTTTAAATGTCGTCATTTCTTGA (5′-3′)
**Nbssn6 (PCR for yeast two-hybrid)**	Forward	CATCGATACGGGATCATGATTTTTTCTAATTTCGCA (5′-3′)
	Reverse	TCATCTGCAGCTCGATCACAAATTTTCCATTTTAAATGTC (5′-3′)
**Nbtrrp2 (PCR for yeast two-hybrid)**	Forward	GAATTCCCGGGGATCATGTACCAACCAGACATTAACAC
	Reverse	GCTAGTTATGCGGCCTTACACCCCTGTCTCTACTGTC
**Nbptp2 (PCR for yeast two-hybrid)**	Forward	GAATTCCCGGGGATCATGTTTTTATCTCTAAACCGAAAAC (5′-3′)
	Reverse	GCTAGTTATGCGGCCTCAAGTAGAATTGGAACCATTTTC (5′-3′)
**Nbssn6 (RNA Oligo)**	Sense	GCCUGGGCCUAUUAUCGAATT (5′-3′)
	Antisense	UUCGAUAAUAGGCCCAGGCTT (5′-3′)
**NC (RNA Oligo)**	Sense	UUCUCCGAACGUGUCACGUTT (5′-3′)
	Antisense	ACGUGACACGUUCGGAGAATT (5′-3′)
**Nbssn6 (RT-qPCR)**	Forward	CAAAGATCCGTTCCTCGTTTATGG (5′-3′)
	Reverse	ATGCGTGTACCATTTTATCGCAATC (5′-3′)
**Nbβ-tubulin (RT-qPCR)**	Forward	TTCCCTTCCCTAGACTTCACTTC (5′-3′)
	Reverse	CAGCAGCCACAGTCAAATACC (5′-3′)
**Nbtrrp2 (RT-qPCR)**	Forward	CCTAGCATAAACTCTAACCCTAGC (5′-3′)
	Reverse	AACCTTCTGGTTCTACGACAAA (5′-3′)
**Nbβ-tubulin (RT-qPCR)**	Forward	AGAACCAGGAACAATGGACG (5′-3′)
	Reverse	AGCCCAATTATTACCAGCACC (5′-3′)

**Table 2 jof-09-00990-t002:** IP results of Nbssn6 by mass spectrometry.

Accession	Description	Coverage	Peptides	PSMs	MW(kD)
EOB15197.1	Heat shock 70 kDa protein 6	33	21	37	73.9
EOB12779.1	Polar tube protein 2	32	7	12	30.9
EOB13492.1	Heat shock protein HSP 90-alpha 1	17	14	19	78.5
EOB12193.1	T-complex protein 1 zeta subunit	28	11	17	44.5
EOB12021.1	T-complex protein 1 subunit eta	21	8	13	51.2
EOB15438.1	T-complex protein 1 subunit delta	21	9	13	55.6
EOB13574.1	T-complex protein 1 subunit epsilon	22	11	17	58.4
EOB11504.1	T-complex protein 1 subunit alpha	16	9	14	59.6
EOB14546.1	T-complex protein 1 subunit theta	7	7	9	113.7
EOB12444.1	T-complex protein 1 subunit beta	14	6	7	57.6
EOB12077.1	Importin subunit alpha-2	11	4	7	60.6
EOB14741.1	Heat shock 70 kDa protein cognate 4	5	4	7	76.4
EOB12136.1	T complex protein 1 theta subunit	10	3	4	31.5
EOB13787.1	Coatomer subunit delta-3	6	3	3	54.1
EOB14835.1	Coatomer subunit beta	4	3	3	86.6

## Data Availability

Not applicable.

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
