# Peer review of "Ssn6 Interacts with Polar Tube Protein 2 and Transcriptional Repressor for RNA Polymerase II: Insight into Its Involvement in the Biological Process of Microsporidium Nosema bombycis"

_jof, 2023, doi:10.3390/jof9100990_

Round 1
Reviewer 1 Report
This work characterized a transcriptional corepressor of microsporidian Nosema bombycis, Nbssn6, mainly by determining its subcellular localizations and interacting proteins, which results provide references for further understanding the functions of this gene in N. bombycis. 1. line 23, "These results suggest that ssn6 may participate in the infection". It is hard to understand that ssn6 may be invovled in the infection according to its subcellular localizations - at nucleus junction in meront stage and cytoplasm in other stages. 2. line 43, full name should be given for the abbreviations where it appears firstly. 3. line 51, should the 'mitochondrial' be mitochondia? 4. lines 75 and 109, database and software papers should be cited. 5. lines 82 and 297, how to understand that Nbssn6 interacts NbPTP2, which locates on the polar tube protein, and at the same time regulates the transcription of Nbptp2, which conducts in the nucleus? This should be explained and discussed. 6. line 91, 'ml' should be written as the mL. 7. lines 96-210, catlog numbers for the reagents and kits used should be provided. 8. lines 115-116 and 204, is the 'qRT-PCR' actually RT-qPCR? How did the authors performed the qPCR? 9. line 175, 'μl' should be written as the μL. 10. chapter 2.7, for a robust result, the authors could exchange the prey and bait vectors to performe another round of the yeast two-hybrid assay. 11. chapter 3.1, how to understand the 3D structers of Nbssn6 (Figure 1C) and NbTRRP2 (Figure 2C)? Both figures should be interpreted in 3.1. The coding regions for the predicted domains in Figure 1B and 2C could be labled in Figure 1D and 2D, respectively. 12. chapter 3.3, it is hard to detect the parasite stages just by DAPI dyeing, but could be accessorily indicated by staining the chitin components. 13. lines 268-269, Figure 4A can not indicate that Nbssn6 locates in the junction between divding nuclei, but shows a cytosol distribution of the Nbssn6. 14. Why there is no NbTRRP2 in the IP results (Table 2)? 15. Regarding to the RNAi analysis, the decrease of Nbssn6, NbTRRP2 and NbPTP2 proteins could be verified using Western blot for a robust result. 16. The biggest problem of this manuscript is that the context and logic among the subcellular locations, interacting proteins, and the proposed functions of Nbssn6 are incompatible and hard to understand. The authors should make it clear.
English language revisions are needed.
Author Response
Manuscript ID: jof-2555348
Title: Ssn6 interacts with polar tube protein 2 and transcriptional re-pressor for RNA polymerase II: insight into its involvement in biological process of microsporidium Nosema bombycis
Dear Editor and reviewers:
Thank you for your letter. We are pleased to know that our work was rated as potentially acceptable for publication in your Journal, subject to adequate revision. We thank the reviewers for the time and effort they have put into reviewing the previous version of the manuscript. Their suggestions have enabled us to improve our work. Based on the instructions provided in your letter, we uploaded the files of the revised manuscript.
Next page of this letter is our point-by-point response to the comments raised by the reviewers. The comments are reproduced, and our revisions are given in different color (red) in the revised manuscript.
We would like also to thank you for allowing us to resubmit a revised copy of the manuscript. We hope that the revised manuscript is accepted for publication by Journal of Fungi.
Sincerely,
Runpeng Wang
Reviewer 1: This work characterized a transcriptional corepressor of microsporidian Nosema bombycis, Nbssn6, mainly by determining its subcellular localizations and interacting proteins, which results provide references for further understanding the functions of this gene in N. bombycis.
Major issues:
- line 23, "These results suggest that ssn6 may participate in the infection". It is hard to understand that ssn6 may be invovled in the infection according to its subcellular localizations at nucleus junction in meront stage and cytoplasm in other stages.
Response: Thank for raising this question. The description in the manuscript was not rigorous. So, we moderated the sentence to “These results suggest that Nbssn6 may impacts on the infection and proliferation of N. bombycis via interacting with polar tube protein and transcriptional repressor for RNA polymerase II.” in line 24. PTP2 is one of the structural proteins of the polar tube in N. bombycis and play an important role in the infection process. ssn6 could interact with PTP2 and downregulate its transcription level. Therefore we suggest ssn6 have connection with the infection process. We also added some relevant literatures about PTP2 in Discussion, lines 437-445.
- line 43, full name should be given for the abbreviations where it appears firstly.
Response: Thank you for pointing out this error, we have modified this mistake in lines 44 in the revised manuscript.
- line 51, should the 'mitochondrial' be mitochondia?
Response: Thank you for pointing out this error, we have replaced “mitochondrial” with “mitochondia” in line 67.
- lines 75 and 109, database and software papers should be cited.
Response: Thank you for pointing out this error. We have cited relevant literatures of every database and software in lines 119-125 of the revised manuscript.
- lines 82 and 297, how to understand that Nbssn6 interacts NbPTP2, which locates on the polar tube protein, and at the same time regulates the transcription of Nbptp2, which conducts in the nucleus? This should be explained and discussed.
Response: This is a very valuable comment. The NbPTP2 protein is synthesized in the ribosome, then assembled into the polar tube protein after modification. So, we further analyzed the distribution of ssn6 during the proliferative phase and sporogonic phase of N. bombycis. As shown in figure6 A, C (lines 311-312 and lines 318-324), besides distributed in the cytoplasm, the ssn6 protein could be found in the nucleus of the meront and sporoblasts as well, which indicates that the ssn6 can shuttle between cytoplasm and nuclei.
- line 91, 'ml' should be written as the mL.
Response: Thank you for pointing out this error. We have modified this mistake in line 101.
- lines 96-210, catlog numbers for the reagents and kits used should be provided.
Response: Thank you for pointing out this error. We checked every reagent and kit and provided the catlog numbers in lines 110-231.
- lines 115-116 and 204, is the 'qRT-PCR' actually RT-qPCR? How did the authors perform the qPCR?
Response: Thank you for pointing out this error and raising this question. The “qRT-PCR” is actually “RT-qPCR” and we have modified throughout the manuscript. The qPCR methods are supplemented and modified in the manuscript in 2.8. RNAi and RT-qPCR, revised as follows:
RT-qPCR was performed according to the TB GreenTM Premix ExTaqTM II (Tli RNase H Plus) (RR820A, Takara Biotechnology, Dalian) kit manufacturer's instructions, conducted at 95°C for 30 s, followed by 40 cycles of 95°C for 5 s and 60°C for 30 s on QuantStudio 3 (ABI, USA). The β-tubulin gene of N. bombycis was used as a reference gene, and the primers could be found in table 1. Three independent repeated experiments were performed. The transcription levels were calculated by the 2-∆∆ct method. GraphPad Prism 8.0 (GraphPad Software, San Diego, CA, United States) was used to conduct the multiple t tests.
- line 175, 'μl' should be written as the μL.
Response: Thank you for pointing out this error. We have modified this mistake in line 189 in the revised manuscript.
- chapter 2.7, for a robust result, the authors could exchange the prey and bait vectors to perform another round of the yeast two-hybrid assay.
Response: Thank for raising kind comment. We have not yet completed this experiment due to the limited time .
- chapter 3.1, how to understand the 3D structers of Nbssn6 (Figure 1C) and NbTRRP2 (Figure 2C)? Both figures should be interpreted in 3.1. The coding regions for the predicted domains in Figure 1B and 2C could be labled in Figure 1D and 2D, respectively.
Response: Thank for raising this question. We remade the images and used red box to mark the domains. We also carefully analyzed the structure of two proteins and added it in chapter 3.1: “Domain analysis shows that the Nbssn6 has four TPRs which are located at 119-152 aa, 156-189 aa, 258-291 aa, and 292-325 aa in the N. bombycis (Fig. 2A). These TPR do-mains were also shown in three-dimensional model, each TPR domain was predicted to form a helix-turn-helix arrangement V-shaped structure (Fig. 2B).” (Lines 239-243); “Domain analysis shows that the TRRP2 contains a SCOP d1gxra domain which is located at 62-293 aa and forms a toroidal structure based on seven blades, each blade consists of a four-stranded antiparallel β sheet (Fig. 3A). The SCOP d1gxra domain shares homology with human Groucho/TLE1 transcriptional corepressor, and is very similar to Tup1 β propellers” (Lines 258-262).
- chapter 3.3, it is hard to detect the parasite stages just by DAPI dyeing, but could be accessorily indicated by staining the chitin components.
Response: This is a very thought-provoking question. In N. bombycis, the outer shell of mature spores are mainly composed of chitin, whereas few chitin exists in the multiplying cells before maturation.
We can distinguish the stages of parasite according the number of nuclei and the shape of cells. The DAPI is used to the nuclei. The cells of proliferative phase is usually multiple nuclei and fusiform in shape, while the cells of sporogonic phase are two or four nuclei and spherical or oval.
- lines 268-269, Figure 4A can not indicate that Nbssn6 locates in the junction between divding nuclei, but shows a cytosol distribution of the Nbssn6.
Response: Thank you for pointing out this error. We have modified this mistake in line 300 in the revised manuscript.
- Why there is no NbTRRP2 in the IP results (Table 2)
Response: Thank for raising this question. The protein samples used for IP assay was mainly from mature spores in which the NbTRRP2, we infer, is very few. Meanwhile, the ssn6-tup1 complex participate in transcriptional regulation which is executed in the early stages of development.
- Regarding to the RNAi analysis, the decrease of Nbssn6, NbTRRP2 and NbPTP2 proteins could be verified using Western blot for a robust result.
Response: Thank for kind comment. It takes several months to prepare an antibody, we have not yet been able to prepare corresponding antibodies after identifying two interacting proteins due to experimental time limitations. There is only ssn6 antibody in our laboratory currently.
- The biggest problem of this manuscript is that the context and logic among the subcellular locations, interacting proteins, and the proposed functions of Nbssn6 are incompatible and hard to understand. The authors should make it clear.
Response: That is an excellent comment. By reanalyzing our results and inferring more literatures, we found that ssn6 exist in the nuclei of meront and sporoblasts where is the main place for transcription, and the TRRP2 is a homologous protein of tup1 and posses similar function with tup1. We have modified the relevant context in the manuscript. Literatures also support TRRP2 as tup1 homologue:
[1] Pickles LM, Roe SM, Hemingway EJ, Stifani S, Pearl LH. Crystal structure of the C-terminal WD40 repeat domain of the human Groucho/TLE1 transcriptional corepressor. Structure. 2002 Jun;10(6):751-61. doi: 10.1016/s0969-2126(02)00768-2. PMID: 12057191.
[2] Flores-Saaib RD, Courey AJ. Analysis of Groucho-histone interactions suggests mechanistic similarities between Groucho- and Tup1-mediated repression. Nucleic Acids Res. 2000 Nov 1;28(21):4189-96. doi: 10.1093/nar/28.21.4189. PMID: 11058116; PMCID: PMC113153.
Reviewer 2 Report
The paper by Wang et al. identified a TPR domain-contained General transcriptional corepressor Nbssn6 and transcriptional repressor for RNA polymerase II NbTRRP2 from microsporidium N. bombycis. In addition, the interaction of Nbssn6 and NbTRRP2 were display by Y2H. The work is important for uncovering the transcriptional regulation mechanism of microsporidia, especially the interacted protein TuP1 in the functional complex is missing in the normal microsporidia. It is with regret that, the author mixed the interaction work of SSN6 with PTP2 and NbTRRP2, which I think is easy to mislead the core question of SSN6. Thus, I suggest that this article only focuses on Nbssn6 and NbTRRP2. Here, I enclose my correction, suggestions and recommendations.
Abstract and Introduction
Line 16, 34: “microsporidian” is an adjective; “microsporidium” is a single noun, please use here and elsewhere; “microsporidia” as a plural, the Microsporidia as a phylum.
The introduction is detailed, while it would be easier to bring in if the origin cause of the study of NbSSN6 could be first mentioned.
I suggest: line 73-75 move to line 46 as ‘We screened the general transcriptional corepressor Nbssn6 in the transcriptome of silkworm midgut infected with microsporidium N. bombycis and the N. bombycis genome database (https://silkpathdb.swu.edu.cn/) according to…. (the reason). Nbssn6 comprises six Tetratricopeptide repeats (TPRs), and each repeat has 34 amino acids. TPR has been found in various proteins in different organisms and play important roles in many essential biological pathways, such as synaptic vesicle fusion, targeting of peroxisome [2,17], importing of mitochondrial and chloroplast [18,19] as well as cell cycle control [20]. Tetratricopeptide motifs exist in the form of independent folds or part of protein folds. ’
Materials and Methods
Line 90: ‘NbTRR2’ might be ‘NbTRRP2’?
Line 104,118: ‘pMD19-T-Nbssn6’ or ‘pMD19-T-nbssn6’? Please keep the description consistent. The same as in line 129 ‘pET-28a- nbssn6’.
Line 151: If the antibody didn’t be purified by affinity chromatography, then ‘Nbssn6 polyclonal antibody’ should be ‘Nbssn6 polyclonal antiserum’.
Line 196: how many silkworms were fed with the mulberry leaves smeared with ? mL of N. bombycis spores? It will help to know about the detail of infection procedure.
Results
Fig.1: I suggest labelling the TPR repeat motif ‘W4-L7-G8-Y11-A20-F24-A27-P32 (doi: 10.1128/IAI.01035-12; doi: 10.1016/s0968-0004(00)89037-4)’ over the TPRs sequence of Nbssn6 in Fig.1 C for better display of protein sequence characteristics. Besides, please check the conserved domains predicted in NCBI, which shows 6 TPRs and some of the ones that did not label in Fig.1 B match the characteristics of the motif (https://www.ncbi.nlm.nih.gov/Structure/cdd/wrpsb.cgi?INPUT_TYPE=live&SEQUENCE=EOB14249.1).
Fig.2: correct: A Lane 1 should be ‘NbTRRP2’.
Fig.3: Please modify the fig format following the instruction of the journal. E.g. crop and rotate the picture properly to make it look more professional and ensure that the font format in the picture is consistent. As for the current version, Times New Roman and Arial are both used.
Fig.4: In my opinion, green signal of Nbssn6 in A, B, C indicates the localization character of the protein in the merogony phase. I suggest you repeat the localization assay by adding the Fluorescent brightener stain in order to better distinguish the different phases of microsporidia according to the shape and thickness of the chitin layer. As I noticed, antibodies were difficult to target the antigen in the cytoplasm of microsporidian sporont, sporoblast and mature spores while in the host cell. Besides, phase in B is more likely the sporoplasm that enters the host.
Fig.5: the same as fig. 3. In addition, have you performed the co-localization of Nbssn6 and NbTRRP2? It will be helpful to know the temporal and spatial conditions for the co-function of the two proteins, especially during the phase that Nbssn6 is located in the nuclei.
3.4: It is better to display the SDS-PAGE of Co-IP for better analysis of interaction candidates.
3.5 and 3.6: Additional experiments like Co-IP need to be performed to confirm the in vivo interaction of the proteins.
Ref. 8: It should be chapter ‘Microsporidia in Insects’ in the book ‘Microsporidia: Pathogens of Opportunity’

Minor editing of English language required.
Author Response
Manuscript ID: jof-2555348
Title: Ssn6 interacts with polar tube protein 2 and transcriptional re-pressor for RNA polymerase II: insight into its involvement in biological process of microsporidium Nosema bombycis
Dear Editor and reviewers:
Thank you for your letter. We are pleased to know that our work was rated as potentially acceptable for publication in your Journal, subject to adequate revision. We thank the reviewers for the time and effort they have put into reviewing the previous version of the manuscript. Their suggestions have enabled us to improve our work. Based on the instructions provided in your letter, we uploaded the files of the revised manuscript.
Next page of this letter is our point-by-point response to the comments raised by the reviewers. The comments are reproduced, and our revisions are given in different color (red) in the revised manuscript.
We would like also to thank you for allowing us to resubmit a revised copy of the manuscript. We hope that the revised manuscript is accepted for publication by Journal of Fungi.
Sincerely,
Runpeng Wang
Reviewer 2: The paper by Wang et al. identified a TPR domain-contained General transcriptional corepressor Nbssn6 and transcriptional repressor for RNA polymerase II NbTRRP2 from microsporidium N. bombycis. In addition, the interaction of Nbssn6 and NbTRRP2 were display by Y2H. The work is important for uncovering the transcriptional regulation mechanism of microsporidia, especially the interacted protein TuP1 in the functional complex is missing in the normal microsporidia. It is with regret that, the author mixed the interaction work of SSN6 with PTP2 and NbTRRP2, which I think is easy to mislead the core question of SSN6. Thus, I suggest that this article only focuses on Nbssn6 and NbTRRP2. Here, I enclose my correction, suggestions and recommendations.
Response: We are very sorry for the reading difficulties. By reanalyzing our results and inferring more literatures, we found that ssn6 exist in the nuclei of meront and sporoblasts where is the main place for transcription, and the TRRP2 is a homologous protein of tup1 and posses similar function with tup1. We have modified the relevant context in the manuscript. Literatures also support TRRP2 as tup1 homologue:
[1] Pickles LM, Roe SM, Hemingway EJ, Stifani S, Pearl LH. Crystal structure of the C-terminal WD40 repeat domain of the human Groucho/TLE1 transcriptional corepressor. Structure. 2002 Jun;10(6):751-61. doi: 10.1016/s0969-2126(02)00768-2. PMID: 12057191.
[2] Flores-Saaib RD, Courey AJ. Analysis of Groucho-histone interactions suggests mechanistic similarities between Groucho- and Tup1-mediated repression. Nucleic Acids Res. 2000 Nov 1;28(21):4189-96. doi: 10.1093/nar/28.21.4189. PMID: 11058116; PMCID: PMC113153.
Major issues:
- Line 16, 34: “microsporidian” is an adjective; “microsporidium” is a single noun, please use here and elsewhere; “microsporidia” as a plural, the Microsporidia as a phylum.
Response: Thank you for pointing out this error, we have modified these errors in title, line 13, 17. We also replaced “microsporidia” by “Microsporidia” in the revised manuscript.
- The introduction is detailed, while it would be easier to bring in if the origin cause of the study of NbSSN6 could be first mentioned.
Response: Thank you for your suggestion. We have supplemented the origin cause of our study in line 66-68: “In order to figure out whether the ssn6 act as transcriptional corepressor in microsporidia. We have screened the database and found that the ssn6 is present in N. bombycis.”
- I suggest: line 73-75 move to line 46 as ‘We screened the general transcriptional corepressor Nbssn6 in the transcriptome of silkworm midgut infected with microsporidium N. bombycis and the N. bombycis genome database (https://silkpathdb.swu.edu.cn/) according to…. (the reason). Nbssn6 comprises six Tetratricopeptide repeats (TPRs), and each repeat has 34 amino acids. TPR has been found in various proteins in different organisms and play important roles in many essential biological pathways, such as synaptic vesicle fusion, targeting of peroxisome [2,17], importing of mitochondrial and chloroplast [18,19] as well as cell cycle control [20]. Tetratricopeptide motifs exist in the form of independent folds or part of protein folds.’
Response: Thank for your suggestions. We have revised our manuscript based on your proposal, and we hope the modified version can satisfy you: “We have screened the general transcriptional corepressor ssn6 in the transcriptome of silkworm midgut infected with N. bombycis and the N. bombycis genome database (https://silkpathdb.swu.edu.cn/)[17]. General transcriptional corepressor ssn6 comprises six TPRs, and each repeat has 34 amino acids. TPR has been found in various proteins in different organisms and play important roles in many essential biological pathways, such as synaptic vesicle fusion, the targeting of peroxisome [2,18], importing of mitochondria and chloroplast [19,20] as well as cell cycle control [21]. Tetratricopeptide motifs exist in the form of independent folds or part of protein folds.” (Line 51-58)
- Line 90: ‘NbTRR2’ might be ‘NbTRRP2’?
Response: Thank you for pointing out this error, we have modified these errors in line 100 in the revised manuscript.
- Line 104,118: ‘pMD19-T-Nbssn6’ or ‘pMD19-T-nbssn6’? Please keep the description consistent. The same as in line 129 ‘pET-28a- nbssn6’.
Response: Thank you for pointing out this error, we have modified these errors in line 130, 141 in the revised manuscript.
- Line 151: If the antibody didn’t be purified by affinity chromatography, then ‘Nbssn6 polyclonal antibody’ should be ‘Nbssn6 polyclonal antiserum’.
Response: Thank for raising this question. The antibody we used was purified by affinity chromatography, and we supplement in line 164-165 in the revised manuscript.
- Line 196: how many silkworms were fed with the mulberry leaves smeared with? mL of N. bombycis spores? It will help to know about the detail of infection procedure.
Response: Thank for raising this question. We have supplemented these details in the revised manuscript: “40 newly molted silkworm larvae of 5th instar were fed on mulberry leaves smeared with 20 mL of N. bombycis spores (108 spores/mL) for 6 h. 3 μL of siRNA (1 μg) or a nonsense fragment was injected into the infected silkworm, respectively. Midgut tissues of 10 silkworm larvae were taken at 24 h, 48 h, 72 h, 96 h, and 120 h post injec-tion, respectively, then washed with PBS and stored at -80 °C.” (Line 210-214). We hope these details will satisfy you.
- Fig.1: I suggest labelling the TPR repeat motif ‘W4-L7-G8-Y11-A20-F24-A27-P32 (doi: 10.1128/IAI.01035-12; doi: 10.1016/s0968-0004(00)89037-4)’ over the TPRs sequence of Nbssn6 in Fig.1 C for better display of protein sequence characteristics. Besides, please check the conserved domains predicted in NCBI, which shows 6 TPRs and some of the ones that did not label in Fig.1 B match the characteristics of the motif (https://www.ncbi.nlm.nih.gov/Structure/cdd/wrpsb.cgi?INPUT_TYPE=live&SEQUENCE=EOB14249.1).
Response: Thank you for your suggestion. The TPR domains was predicted by SMART (http://smart.embl-heidelberg.de/) in the former manuscript. However, there are six TPRs in Nbssn6 predicated in NCBI. After using multiple tools and statistical analysis again, we found that there are four TPRs that was located at 119-152 aa, 156-189 aa, 258-291 aa, and 292-325 aa, with high degree of conservation, while other two TPRs that was located at 50-78 aa and 190-218 aa, with relatively low conservation, which may result from distant genetic relationship between the Microsporidia and other species. Therefore, we remade the Figure 2 and marked the domains on the three-dimensional model and sequence. TPRs predicted in NCBI database were marked in red.
- Fig.2: correct: A Lane 1 should be ‘NbTRRP2’.
Response: Thank you for pointing out this error, we have recombined the images and modified the agarose gel image. The modified image could be found in line 247-249 in the revised manuscript.
- Fig.3: Please modify the fig format following the instruction of the journal. E.g. crop and rotate the picture properly to make it look more professional and ensure that the font format in the picture is consistent. As for the current version, Times New Roman and Arial are both used.
Response: Thank you for pointing out this error. The remade figure is shown in line 283 in the revised manuscript.
- Fig.4: In my opinion, green signal of Nbssn6 in A, B, C indicates the localization character of the protein in the merogony phase. I suggest you repeat the localization assay by adding the Fluorescent brightener stain in order to better distinguish the different phases of microsporidia according to the shape and thickness of the chitin layer. As I noticed, antibodies were difficult to target the antigen in the cytoplasm of microsporidian sporont, sporoblast and mature spores while in the host cell. Besides, phase in B is more likely the sporoplasm that enters the host.
Response: This is a very thought-provoking question. In N. bombycis, the outer shell of mature spores are mainly composed of chitin, whereas few chitin exists in the multiplying cells before maturation.
We can distinguish the stages of parasite according the number of nuclei and the shape of cells. The DAPI is used to the nuclei. The cells of proliferative phase is usually multiple nuclei and fusiform in shape, while the cells of sporogonic phase are two or four nuclei and spherical or oval.
- Fig.5: the same as fig. 3. In addition, have you performed the co-localization of Nbssn6 and NbTRRP2? It will be helpful to know the temporal and spatial conditions for the co-function of the two proteins, especially during the phase that Nbssn6 is located in the nuclei.
Response: Thank for kind comment. It takes several months to prepare an antibody, we have not yet been able to prepare corresponding antibodies after identifying two interacting proteins due to experimental time limitations. There is only ssn6 antibody in our laboratory currently.
- 3.4: It is better to display the SDS-PAGE of Co-IP for better analysis of interaction candidates.
Response: Thank for kind suggestion. We have put the SDS-PAGE of Co-IP into the revised manuscript in line 330-334.
- 3.5 and 3.6: Additional experiments like Co-IP need to be performed to confirm the in vivo interaction of the proteins.
Response: Thank for raising this question. The protein samples used for Co-IP assay was mainly from mature spores in which the NbTRRP2, we infer, is very few. Meanwhile, the ssn6-tup1 complex participate in transcriptional regulation which is executed in the early stages of development.
- Ref. 8: It should be chapter ‘Microsporidia in Insects’ in the book ‘Microsporidia: Pathogens of Opportunity’
Response: Thank you for pointing out this error. We have modified the format of books to fit the requirements of this journal: “Becnel, James J., and Theodore G. Andreadis. Microsporidia in Insects. In Book The Microsporidia and Microsporidiosis, 2nd ed.; M. Wittner and L.M. Weiss, Eds.; USA, 2014; 14, pp. 521-570” (Line 477-478).
Reviewer 3 Report
Microsporidia are obligate intracellular parasites that cause disease in many commercially important invertebrates. However, we understand relatively little about their biology compared to other microbial pathogens. Here, the authors attempt to explore the role of a putative Ssn6 homolog in the biology of the microsporidia species Nosema bombycis which is the oldest known microsporidia species that causes disease in silkworms. This is an important area of research in microsporidia as our understanding of their cell regulatory processes is quite limited. The authors are well positioned in expertise and experience to carry out the work based on their previous work. The authors identify the Nb Ssn6 homolog, look at its cellular localization, and show binding of Ssn6 with other proteins including the polar tube protein PTP2. Finally, the authors show that knockdown of the Ssn6 reduces its expression and leads to changes in expression of PTP2. Although the manuscript details some very interesting findings, the work is marked by some issues which make it unsuitable for publication in its present form. The primary concern focuses on the number of experimental repeats performed. The second issue centers on critical data about how Ssn6 knockdown impacts infection. A third concern centers on the inclusion of TRRP2 in this manuscript as it does not seem particularly relevant to the study based on the provided information. In addition, the manuscript would be much stronger if 1) the authors develop and present a model for how Ssn6 impacts PTP2 expression and why the two proteins interact, and 2) the authors address the dominance of chaperones and other protein folding machinery in the Mass spec data for Ssn6. Finally, the manuscript could use are some writing improvements to better communicate the findings to the reader.
Comments:
A serious concern focuses on the number of experimental repeats performed. The authors must include information about how many independent times they performed each experiment.
A second critical point to establish is how RNAi-mediated reduced Ssn6 expression impacts N. bombycis levels. This question is important for understanding how Ssn6 fits into N. bombycis biology and it should be pretty straightforward for them to complete. The authors already have a very powerful loss-of-function system with RNAi. They could measure infection levels in a timecourse using RNA they have already collected to examine levels of a housekeeping gene, using DNA from an infection to measure genome equivalents, or using microscopic methods to visualize infection in order to answer this important question.
A third issue is that the inclusion of the NbTRRP2 experiments in the manuscript does not add much and they should probably be taken out for addition to a paper detailing more about this protein’s function. At the very least, a better explanation of why it is included and the limitations of the experiments involving it should be undertaken. It appears that the authors decide to look at NbTRRP2 solely because this protein “has been annotated as transcriptional repressor for RNA polymerase II”. This rationale is not compelling for studying the protein, especially as the name in the database seems completely arbitrary as the protein has no demonstrated homology to other known proteins. Homology to Groucho/TLE1 is mentioned in the text but not supported by evidence. While the authors do find the protein interacts with Ssn6 using yeast-2-hybrid technique, NbTRRP2 was not found in the Mass Spec results from their more biologically relevant CoIP, making the interaction seem less germane to this paper
There are a number of changes that would make the manuscript much stronger.
· First, the authors should develop and present a model for how Ssn6 impacts PTP2 expression and why the two proteins interact This especially important in light of the differential effetcs of Ssn6 knockdown on PTP2 expression at different timepoints.
· Second, the authors address the dominance of chaperones and other protein folding machinery in the Mass spec data for Ssn6. Why didn't they follow this up?
there are some typos and sections that could use spelling and grammer improvements to better communicate the findings to the reader. Some examples (not a complete list) follow. In the abstract, the second sentence (Ln 13 and 14) needs reworking to make sense. In the introduction, the sentence that starts on Ln 42 also needs improvement. In Figure 2A, the cloned fragment for NbTRRP2 appears to be mislabeled as NbPTP2.
Author Response
Manuscript ID: jof-2555348
Title: Ssn6 interacts with polar tube protein 2 and transcriptional re-pressor for RNA polymerase II: insight into its involvement in biological process of microsporidium Nosema bombycis
Dear Editor and reviewers:
Thank you for your letter. We are pleased to know that our work was rated as potentially acceptable for publication in your Journal, subject to adequate revision. We thank the reviewers for the time and effort they have put into reviewing the previous version of the manuscript. Their suggestions have enabled us to improve our work. Based on the instructions provided in your letter, we uploaded the files of the revised manuscript.
Next page of this letter is our point-by-point response to the comments raised by the reviewers. The comments are reproduced, and our revisions are given in different color (red) in the revised manuscript.
We would like also to thank you for allowing us to resubmit a revised copy of the manuscript. We hope that the revised manuscript is accepted for publication by Journal of Fungi.
Sincerely,
Runpeng Wang
Reviewer 3: Microsporidia are obligate intracellular parasites that cause disease in many commercially important invertebrates. However, we understand relatively little about their biology compared to other microbial pathogens. Here, the authors attempt to explore the role of a putative Ssn6 homolog in the biology of the microsporidia species Nosema bombycis which is the oldest known microsporidia species that causes disease in silkworms. This is an important area of research in microsporidia as our understanding of their cell regulatory processes is quite limited. The authors are well positioned in expertise and experience to carry out the work based on their previous work. The authors identify the Nb Ssn6 homolog, look at its cellular localization, and show binding of Ssn6 with other proteins including the polar tube protein PTP2. Finally, the authors show that knockdown of the Ssn6 reduces its expression and leads to changes in expression of PTP2. Although the manuscript details some very interesting findings, the work is marked by some issues which make it unsuitable for publication in its present form. The primary concern focuses on the number of experimental repeats performed. The second issue centers on critical data about how Ssn6 knockdown impacts infection. A third concern centers on the inclusion of TRRP2 in this manuscript as it does not seem particularly relevant to the study based on the provided information. In addition, the manuscript would be much stronger if 1) the authors develop and present a model for how Ssn6 impacts PTP2 expression and why the two proteins interact, and 2) the authors address the dominance of chaperones and other protein folding machinery in the Mass spec data for Ssn6. Finally, the manuscript could use some writing improvements to better communicate the findings to the reader.
Major issues:
- A serious concern focuses on the number of experimental repeats performed. The authors must include information about how many independent times they performed each experiment.
Response: Thank you for your comment. We conducted Co-IP for more than three times, and chose the SDS-PAGE with most obvious specific bands for further mass spectrometry analysis. Immunolocalization was also repeated more than three times until satisfactory results. In Yeast-two hybrid assay, we selected more than five positive clones on SD−Leu/−Trp plates that was selected and cultured on SD−Ade/−His/−Leu/−Trp plates con-taining X-α-gal. After 3 days, three clones that turned blue were considered as positive results. cDNA used for RNAi and RT-qPCR was derived from at least five silkworms. Every result of RT-qPCR for statistics was made up of at least three biological samples.
- A second critical point to establish is how RNAi-mediated reduced Ssn6 expression impacts bombycis levels. This question is important for understanding how Ssn6 fits into N. bombycis biology and it should be pretty straightforward for them to complete. The authors already have a very powerful loss-of-function system with RNAi. They could measure infection levels in a timecourse using RNA they have already collected to examine levels of a housekeeping gene, using DNA from an infection to measure genome equivalents, or using microscopic methods to visualize infection in order to answer this important question.
Response: Thank you for your comment. We have established the standard curve of Nbβ -tublin and calculated the copy number of housekeeping gene Nbβ -tublin at different time points of the NC and RNAi samples. As a result, we found the proliferation level was significantly downregulated at 24h, and very significant at 72h and 96h. These results indicated that Ssn6 is involved in the infection and proliferation of the N. bombycis. We have added relevant context in the revised manuscript.
- A third issue is that the inclusion of the NbTRRP2 experiments in the manuscript does not add much and they should probably be taken out for addition to a paper detailing more about this protein’s function. At the very least, a better explanation of why it is included and the limitations of the experiments involving it should be undertaken. It appears that the authors decide to look at NbTRRP2 solely because this protein “has been annotated as transcriptional repressor for RNA polymerase II”. This rationale is not compelling for studying the protein, especially as the name in the database seems completely arbitrary as the protein has no demonstrated homology to other known proteins. Homology to Groucho/TLE1 is mentioned in the text but not supported by evidence. While the authors do find the protein interacts with Ssn6 using yeast-2-hybrid technique, NbTRRP2 was not found in the Mass Spec results from their more biologically relevant CoIP, making the interaction seem less germane to this paper
Response: That is an excellent comment. By reanalyzing our results and inferring more literatures, we found that ssn6 exist in the nuclei of meront and sporoblasts where is the main place for transcription, and the TRRP2 is a homologous protein of tup1 and posses similar function with tup1. We have modified the relevant context in the manuscript. Literatures also support TRRP2 as tup1 homologue:
[1] Pickles LM, Roe SM, Hemingway EJ, Stifani S, Pearl LH. Crystal structure of the C-terminal WD40 repeat domain of the human Groucho/TLE1 transcriptional corepressor. Structure. 2002 Jun;10(6):751-61. doi: 10.1016/s0969-2126(02)00768-2. PMID: 12057191.
[2] Flores-Saaib RD, Courey AJ. Analysis of Groucho-histone interactions suggests mechanistic similarities between Groucho- and Tup1-mediated repression. Nucleic Acids Res. 2000 Nov 1;28(21):4189-96. doi: 10.1093/nar/28.21.4189. PMID: 11058116; PMCID: PMC113153.
- The authors should develop and present a model for how Ssn6 impacts PTP2 expression and why the two proteins interact. This especially important in light of the differential effects of Ssn6 knockdown on PTP2 expression at different time points.
Response: Thank you for raising this propose. TRRP2 has similar domain with transcriptional repressors of the Groucho/transducin-like Enhancer of split (TLE) family. Proteins with similar domains in this family has similar function with tup1 and could interact with ssn6. PTP2 is the constituent protein of the polar tube. The PTP1-PTP2 complex has been proved to play important role in the infection process of Microsporidia. The ssn6 interact with TRRP2 and PTP2, and regulate the transcription of TRRP2 and PTP2, suggesting that the ssn6 effect the infection of N. bombycis via forming the ssn6-TRRP2 complex.
- The authors address the dominance of chaperones and other protein folding machinery in the Mass spec data for Ssn6. Why didn't they follow this up?
Response: In the numerous mass spectrometry results, we selected some protein with high rank and conducted the Yeast-two hybrid assay. Unfortunately, only the PTP2 showed positive results on SD−Ade/−His/−Leu/−Trp plates. In other species, ssn6 usually work together with tup1, although we failed to search out the protein named as tup1 of N. bombycis in the database, we found the TRRP2 by blast using the homologous sequence for retrieval. Domains analyses showed that the TRRP2 has similar function with tup1, so we focused on the TRRP2 in our subsequent experiments.
- there are some typos and sections that could use spelling and grammar improvements to better communicate the findings to the reader. Some examples (not a complete list) follow. In the abstract, the second sentence (Ln 13 and 14) needs reworking to make sense. In the introduction, the sentence that starts on Ln 42 also needs improvement. In Figure 2A, the cloned fragment for NbTRRP2 appears to be mislabeled as NbPTP2.
Response: We are sorry for the reading difficulties. We have modified the sentences you mentioned in line 13,14 and line 42. The mislabeled NbTRRP2 has been corrected as NbPTP2. We also carefully checked spelling and grammar throughout the manuscript and modified the writing that may have grammar or other problems in the revised manuscript.
Round 2
Reviewer 2 Report
Line 45-47: ‘whereas the all enzymes participating in glycolysis and part of the enzymes in pentose phosphate pathway has been identified’ Actually, parts of enzymes participating in glycolysis were also lost in some species of Enterocytozoonidae, e. g. Enterocytozoon bieneusi, E. hepatopenaei, Enterospora canceri (doi: 10.1111/1462-2920.13734).
Line 206, 208: The optimum temperature for yeast is 30 ℃, please confirm the actual growth temperature or a clerical error.
Fig. 5 and 6: I agree that stages of microsporidia can be distinguished by the number of nuclei and the shape of the cell and nuclei. In Fig. 5 A &6A, most microsporidia cells are meronts, while the pointed ones are merons in division. The white arrow indicated elongating meront, and the white arrowhead indicated meront is ready for division. Fig. 5B, the parasite cells were supposed to be sporoplasm or early phase of meront. Because these cells are spherical and cannot be stained by fluorescent brightener in our research, thus they are probably not sporonts. Fig. 5C & 6C, the chitin layer of microsporidia has already formed in sporoblast and the sharp are more likely mature spores, they can be seen in light view of the host cell. Besides, the nuclei of sporoblast are smaller and more compact. Therefore, this should still be the meronts. Because the work focused on SSN6 during the proliferative phase, the lack of localization in sporongonic phase will not influence the main conclusions of the article.
It can be read smoothly.
Author Response
Manuscript ID: jof-2555348
Title: Ssn6 interacts with polar tube protein 2 and transcriptional repressor for RNA polymerase II: insight into its involvement in biological process of microsporidium Nosema bombycis
Dear reviewer:
Thank you for your suggestions for revising our manuscript. We have carefully revised the issue you mentioned, we hope the revised manuscript can meet your satisfaction.
Next page of this letter is our point-by-point response to the comments. The comments are reproduced, and our revisions are given in different color (red) in the revised manuscript.
Sincerely,
Runpeng Wang
Reviewer 2:
Major issues:
- Line 45-47: ‘whereas the all enzymes participating in glycolysis and part of the enzymes in pentose phosphate pathway has been identified’ Actually, parts of enzymes participating in glycolysis were also lost in some species of Enterocytozoonidae, e. g. Enterocytozoon bieneusi, E. hepatopenaei, Enterospora canceri (doi: 10.1111/1462-2920.13734).
Response: Thank you for your comment. We have changed the related description and cited the reference you mentioned in the revised manuscript. Besides, we also added some other content to make the introduction more logical in lines 48-53.
- Line 206, 208: The optimum temperature for yeast is 30 ℃, please confirm the actual growth temperature or a clerical error.
Response: Thank you for pointing out this error, we have modified these errors in lines 213-214 in the revised manuscript.
- 5 and 6: I agree that stages of microsporidia can be distinguished by the number of nuclei and the shape of the cell and nuclei. In Fig. 5 A &6A, most microsporidia cells are meronts, while the pointed ones are merons in division. The white arrow indicated elongating meront, and the white arrowhead indicated meront is ready for division. Fig. 5B, the parasite cells were supposed to be sporoplasm or early phase of meront. Because these cells are spherical and cannot be stained by fluorescent brightener in our research, thus they are probably not sporonts. Fig. 5C & 6C, the chitin layer of microsporidia has already formed in sporoblast and the sharp are more likely mature spores, they can be seen in light view of the host cell. Besides, the nuclei of sporoblast are smaller and more compact. Therefore, this should still be the meronts. Because the work focused on SSN6 during the proliferative phase, the lack of localization in sporongonic phase will not influence the main conclusions of the article.
Response:Thank you for your careful analysis and proposal. We further analyzed the fluorescence signals and remade the Figure 5 and Figure 6. The results showed that the Ssn6 was mainly distributed in the nuclei and cytoplasm, and can shuttle between the nuclei and cytoplasm at the proliferative phase. We also modified the description about the developmental stage of N. bombycis in lines 306-344.

Reviewer 3 Report
I appreciate the efforts of the authors to address my concerns and believe the paper is now ready for publication in its improved form.
Author Response
Thank you for your affirmation of our work, we are glad that you are satisfied with our work.